# AN ANALYSIS OF THE CAUCHY METHOD FOR DIFFERENT STEPLENGTH COEFFICIENT CONFERENCE SUBMISSIONS

## ABSTRACT

In this work we take the parameter r (recipprocal of optimal steplenth) as analysis target and introduce steplength coefficient t for classical steepest descent method for convex quadratic optimization problems, and we found the different coefficients affect the state of the entire system convergence. As the value of t varies, the overall system, including the value of r, may converge towards a fixed value, oscillate between two regions, or display chaotic behavior. We also conducted a specific analysis in the two-dimensional case.

## 1 INTRODUCTION

In this paper, we consider the the unconstrained optimization problem with convex quadratic form

$$minf(x) = \frac{1}{2}x^T A x - b^T x \tag{1}$$

where $x \in \mathbb{R}^n, b \in \mathbb{R}^n, A \in \mathbb{R}^{n \times n}$ is a symmetric and positive definite matrix.

The common solution methods for solving Eq(1) are iterarive methods of the following form

$$x_{k+1} = x_k - \alpha_k \nabla f(x_k) \tag{2}$$

where $\alpha_k$ is a steplenth,gradient descent method and its variants are the most common optimization method.for GD method,if we minimizes Eq.(3) with exact line search,then we get

$$\alpha_k^{SD} = \frac{\nabla f_k^T \nabla f_k}{\nabla f_k^T A \nabla f_k} = \frac{g_k^T g_k}{g_k^T A g_k} \tag{3}$$

$$r_k = \frac{1}{2\alpha_k} = \frac{g_k^T A g_k}{2g_k^T g_k} \tag{4}$$

this method proposed by A.Cauchy (1847) is called steepest descent method ,so $\alpha_k^{SD}$ is also called Cauchy step length. the method's convergence rate is very sensitive to ill condition number and may be very slow ,when the f(x) is quadratic $x_k$ will satisfy the

$$\frac{f(x_{k+1}) - f(x^*)}{f(x_k) - f(x^*)} \leq (\frac{\lambda_1 - \lambda_n}{\lambda_1 + \lambda_n})^2 \tag{5}$$

The convergence rate of SD method is relatively slow with a zigzag phenomena which is proved by Akaike (1959) and Forsythe (1968)

Yuan (2006) proposed a new stepsize formula for the SD method. the method alternates as follows, on even iterations it employs the Yuan step size, while on odd iterations it performs an exact line search. For two-dimensional convex quadratic functions, this alternating scheme guarantees convergence to the minimum in only three iterations. Yuan steplength is as follows:

$$\alpha_k^Y = \frac{2}{\sqrt{\left(\frac{1}{\alpha_{k-1}^{SD}} - \frac{1}{\alpha_k^{SD}}\right)^2 + 4\frac{\|g_{2k}\|^2}{(\alpha_{k-1}^{SD}\|g_{k-1}\|)^2}} + \frac{1}{\alpha_{k-1}^{SD}} + \frac{1}{\alpha_k^{SD}}} \tag{6}$$

Raydan M (2002) proposed RSD which accelerates convergence by introducing a relaxation parameter between 0 and 2 in the standard SD method,In each iteration, the step size is randomly chosen from a fixed interval in $[0, 2\alpha_K^{SD}]$,this randomization eliminates the oscillations inherent in the SD method. RSD method convergs monotonically to optimal point $x^*$ Serafino;F.Riccio;G.Toraldo (2013) observes that over-relaxation appears more suitable than under-relaxation of the Cauchy step size,therefore, they introduces a modified version of RSD, called RSDA method, where $\alpha_k \in [0.8\alpha_k, 2\alpha_k]$.

Kalousek (2015) presents a randomized steepest descent method for minimizing smooth functions. Instead of using exact step sizes, it randomly selects step lengths from a specific probability distribution,where $\alpha_k \in [\frac{1}{\lambda_1}, \frac{1}{\lambda_n}]$

In this paper,we take the parameter $r$ (Eq.(5)) as analysis target and introduce a multiplicative factor parameter $s$ to the SD method and analyze how different values of $s$ affect the method. its formula is as follows:

$$x_{k+1} = x_k - s\alpha_k^{SD}\nabla f(x_k) \tag{7}$$

the same conclusion can be obtained by comparing the simplest form of qudratic function and matrix form. For the convenience of analysis and greater intuitiveness,consider a situation the objective function is a simple n dimensions hyper-ellipsoid stimulating Eq(1)

$$f(x) = \sum_{i=1}^n a^{(i)}x^{(i)^2} \tag{8}$$

$$r = \frac{\sum_{i=1}^n a^{(i)^3}x^{(i)^2}}{\sum_{i=0}^n a^{(i)^2}x^{(i)^2}} = \frac{\sum_{i=1}^n a^{(i)}g^{(i)^2}}{\sum_{i=1}^n g^{(i)^2}} \tag{9}$$

where $0 < a^{(n)} \le a^{(n-1)} \le ...... \le a^{(1)}$,$g^{(i)} = 2a^{(i)}x^{(i)}$, the initial point $X_0 = [x_0^{(1)}, x_0^{(2)}, ......x_0^{(n)}]$ we have

$$r_k = \frac{\sum_{i=1}^n a^{(i)}g_k^{(i)^2}}{\sum_{i=1}^n g_k^{(i)^2}} \tag{10}$$

$$r_{k+1} = \frac{\sum_{i=1}^n a^{(i)}g_k^{(i)^2}(r_k - a^{(i)})^2}{\sum_{i=1}^n a^{(i)}g_k^{(i)^2}(r_k - a^{(i)})^2} \tag{11}$$

now from Eq(7) then we have

$$x_{k+1} = x_k - s\alpha_k^{SD}\nabla f(x_k) = x_k - \frac{\nabla f(x_k)}{tr_k} \tag{12}$$

where $s > 0, s = \frac{1}{t}$

$$r_{k+1} = \frac{\sum_{i=1}^n a^{(i)}g_k^{(i)^2}(tr_k - a^{(i)})^2}{\sum_{i=1}^n a^{(i)}g_k^{(i)^2}(tr_k - a^{(i)})^2} \tag{13}$$

$$r_{k+1} = G(r_k) \tag{14}$$

we will study the functional relationship of $G$ and the effect of parameters $t$ on the function $G$.

## 2  TWO DIMENSION

In two dimensions case, we can analyse explicitly the positive quadratic case.

from Eq(11)

$$r_{k+1} = \frac{a^{(1)}g_k^{(1)^2}(tr_k - a^{(1)})^2 + a^{(2)}g_k^{(2)^2}(tr_k - a^{(2)})^2}{g_k^{(1)^2}(tr_k - a^{(1)})^2 + g_k^{(2)^2}(tr_k - a^{(2)})^2} \tag{15}$$

we treat $r_k$ as a continuous variable $r$,we have

$$G(r) = \frac{a^{(1)}(r - a^{(2)})(tr - a^{(1)})^2 - a^{(2)}(r - a^{(1)})(tr - a^{(2)})^2}{(tr - a^{(1)})^2(r - a^{(2)}) - (tr - a^{(2)})^2(r - a^{(1)})} \tag{16}$$

where $r \in (a^{(2)}, a^{(1)}), G(r) \in (a^{(2)}, a^{(1)})$,differentiate the function $G(r)$

$$G(r)' = (tr - a^{(1)})(tr - a^{(2)})(a^{(1)} - a^{(2)})^2$$
$$\times \frac{(tr - a^{(1)})(tr - a^{(2)}) - 2t(r - a^{(1)})(r - a^{(2)})}{[(tr - a^{(1)})^2(r - a^{(2)}) - (tr - a^{(2)})^2(r - a^{(1)})]^2} \tag{17}$$

if we set $G(r)'$ to zero,we can obtain four solutions of $G(r)'$:

$$r_1 = \frac{a^{(1)}}{t} \tag{18}$$

$$r_2 = \frac{a^{(2)}}{t} \tag{19}$$

$$r_3 = \frac{a^{(1)} + a^{(2)}}{2(2 - t)} - \frac{\sqrt{t^2(a^{(1)} + a^{(2)})^2 - 4t(2 - t)(2t - 1)a^{(1)}a^{(2)}}}{2t(2 - t)} \tag{20}$$

$$r_4 = \frac{a^{(1)} + a^{(2)}}{2(2 - t)} + \frac{\sqrt{t^2(a^{(1)} + a^{(2)})^2 - 4t(2 - t)(2t - 1)a^{(1)}a^{(2)}}}{2t(2 - t)} \tag{21}$$

we can find fixed points $r_e(r_e = G(r_e))$obviously.

$$r_e = \frac{a^{(1)} + a^{(2)}}{2t} \tag{22}$$

put the $r_e$ into Eq(17)

$$G(r_e)' = 1 + \frac{2t(r_e - a^{(1)})(r_e - a^{(2)})}{(\frac{a^{(1)} - a^{(2)}}{2})^2}$$
$$= 1 - \frac{8(ta^{(1)}a^{(2)} + \frac{(\frac{a^{(1)} + a^{(2)}}{2})^2}{t} - \frac{(a^{(1)} + a^{(2)})^2}{2})}{(a^{(1)} - a^{(2)})^2} \tag{23}$$

We will discuss three situations based on the different values of $t$.

## 2.1 $t > 1$

because $r_2 < a^{(2)}, r_4 > a^{(1)}$,so $r_2$ and $r_4$ are out of range. $G(r_e)^{'}$ is a monotony decrease function of t value.when $t$ approach 1, $G(r_e)^{'}$ reach its maximum with $-1$. so $G(r_e)^{'} < -1$. that means $r_e$ is repulsion point. the $r$ value is a chaos motion.from Eq(18),if $r$ approach $a^{(1)}$,$G(r)$ will also approach $a^{(1)}$ , so $r_e = a^{(1)}$ is also a fixed point

$$G(r_e)^{'} = \frac{ta^{(1)} - a^{(2)}}{ta^{(1)} - a^{(1)}} \approx \frac{t}{t-1} > 1 \tag{24}$$

so $r_e = a^{(1)}$ is also repulsion point. From Figure(1a), it can be seen that the function graphs are similar for different values of t. In Figure(1b), the intersection points of the three functions $G(r)$,The inverse function$G(r)^{-1}$ of $G(r)$, and $Y = Y(x)$ are the fixed points. It is evident that the gradient at the fixed point forms an angle less than 90 degrees with Y, indicating that it is a repulsion point.

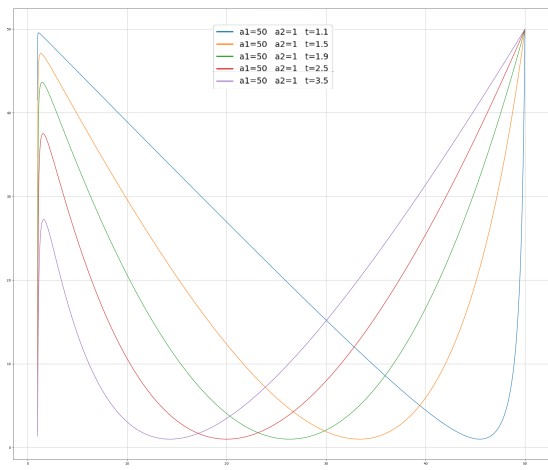

(a) $G(r)$ function $(t = 1.1, 1.5, 1.9, 2.5, 3.5)$

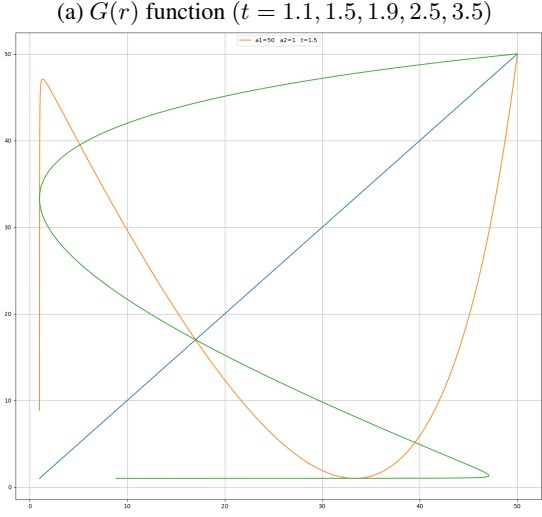

(b) $G(r)$ is orange line ,$G(r)^{-1}$ is green line,$Y(x) = x$ (t=1.5) is blue line

Figure 1: $G(r) function(a^{(1)} = 50, a^{(2)} = 1)$

## 2.2 $t = 1$

Obviously,when $t = 1$,it is the most commonly used steepest descent method.

the initial point $X_0 = [x_0^{(1)}, x_0^{(2)}]$,

$$r_0 = \frac{a^{(1)^3} x_0^{(1)^2} + a^{(2)^3} x_0^{(2)^2}}{a^{(1)^2} x_0^{(1)^2} + a^{(2)^2} x_0^{(2)^2}} = \frac{a^{(1)} g_0^{(1)^2} + a^{(2)} g_0^{(2)^2}}{g_0^{(1)^2} + g_0^{(2)^2}} \tag{25}$$

$$r_1 = \frac{a^{(1)} g_0^{(1)^2} (r_0 - a^{(1)})^2 + a^{(2)} g_0^{(2)^2} (r_0 - a^{(2)})^2}{g_0^{(1)^2} (r_0 - a^{(1)})^2 + g_0^{(2)^2} (r_0 - a^{(2)})^2} = \frac{a^{(2)} g_0^{(1)^2} + a^{(1)} g_0^{(2)^2}}{g_0^{(1)^2} + g_0^{(2)^2}} \tag{26}$$

$$r_2 = \frac{a^{(1)} g_1^{(1)^2} (r_1 - a^{(1)})^2 + a^{(2)} g_1^{(2)^2} (r_1 - a^{(2)})^2}{g_1^{(1)^2} (r_1 - a^{(1)})^2 + g_1^{(2)^2} (r_1 - a^{(2)})^2} = \frac{a^{(2)} g_1^{(1)^2} + a^{(1)} g_1^{(2)^2}}{g_1^{(1)^2} + g_1^{(2)^2}} \tag{27}$$

so

$$r_0 = r_{2k}, r_1 = r_{2k+1} \tag{28}$$

$$r_0 + r_1 = r_k + r_{k+1} = a^{(1)} + a^{(2)} \tag{29}$$

in two dimensions, $r$ will immediately achieve stable state, and then alternate between two values, one large and one small.

from the previous chapter, we know $r_e = \frac{a^{(1)} + a^{(2)}}{2}$, $G(r)' = -1$. Therefore, $r_e$ is a critical state, meaning it is neither attractive nor repulsive. As analyzed earlier, it alternates between the two states.

### 2.3 $t < 1$

It may be concluded that $t > \frac{a^{(1)} + a^{(2)}}{2a^{(1)}}$, If the $t$ value is limited in the interval of $(0.5 + 0.5 \frac{a^{(2)}}{a^{(1)}}, 1)$, $|G(r_e)'| < 1$, the point $r_e$ is a strange attractor, so the $r$ value will tend to the point of $r_e$ when $r$ approach $a^{(1)}$ then

$$G(r_e)' = \frac{ta^{(1)} - a^{(2)}}{ta^{(1)} - a^{(1)}} \approx \frac{t}{t - 1} < -1 \tag{30}$$

so $r_e = a^{(1)}$ is also repulsion point. when $t$ value has been smaller which means $\frac{a^{(1)} + a^{(2)}}{2t} > a^{(1)}$, that means $t < 0.5 + 0.5 \frac{a^{(2)}}{a^{(1)}}$ so in the field only includes one equilibrium point which $r \approx a^{(1)}$,

$$-1 < G(r_e)' < 0 \tag{31}$$

,the point $a^{(1)}$ is a strange attractor, so the r value will tend to the point of $a^{(1)}$ So if $t$ is within a certain range, the value of $r$ is proportional to $t$. However, if $t$ exceeds this range, then $t$ approaches the maximum value of the eigenvector $a^{(1)}$.

## 3 N DIMENSION

Similarly to the previous chapter, for the N-dimensional case, we also conduct an analysis based on different values of $t$.

### 3.1 $t = 1$

when the case t value equal to 1. that means SD methods. Akaike (1959) and Forsythe (1968) have conducted an in-depth analysis, and we have analyzed it from the perspective of $r$. from Eqs(10) and (11)

$$r_k + r_{k+1} = \frac{\sum_{i=1}^{n}\sum_{j=1}^{n} g_k^{(i)^2} g_k^{(j)^2} A(a^{(i)}, a^{(j)})}{\sum_{i=1}^{n}\sum_{j=1}^{n} g_k^{(i)^2} g_k^{(j)^2} B(a^{(i)}, a^{(j)})} \quad (32)$$

where

$$A(x,y) = (x-y)^2(x+y) \quad (33)$$
$$B(x,y) = (x-y)^2 \quad (34)$$

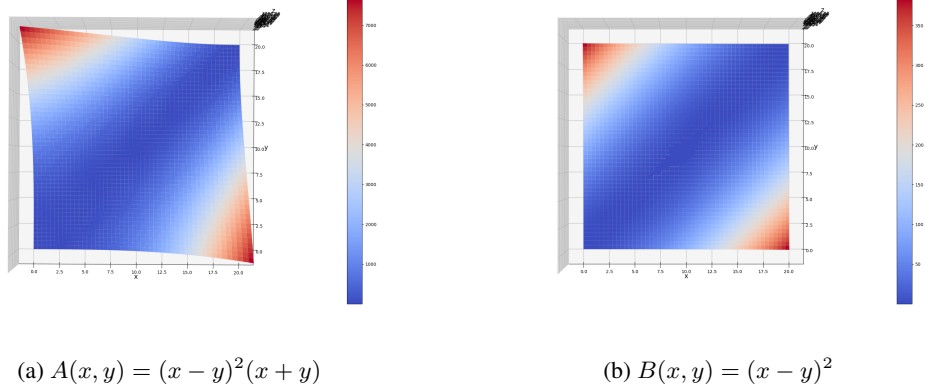

(a) $A(x,y) = (x-y)^2(x+y)$        (b) $B(x,y) = (x-y)^2$

Figure 2: A(x,y) and B(x,y) $0.1 \leq x \leq 20, 0.1 \leq y \leq 20$

then we can see $A(a^{(i)}, a^{(j)})$ and $B(a^{(i)}, a^{(j)})$ as the different weight of the numerator and denominator of Eq(32).

so the bigger the difference between the $a^{(i)}$ and $a^{(j)}$, the greater the weight in $a^{(i)}$ and $a^{(j)}$, from Figure 2, the x and y more center at the top left corner area and the bottom right corner area. the x and y in other areas lead to critical value of A and B,so only the $a^{(i)}$ and $a^{(j)}$ locate in the maximum eigenvector direction area apporximate $a^{(1)}$ and the minimum eigenvector direciton area apporximate $a^{(n)}$ have the biggest weight. Based on the analysis above, Eq(32) is mainly affected by the value at maximum eigenvalue ares and minimum eigenvalue area. after a few step, the system will fall into a state of balance situation,

$$r_k + r_{k+1} \approx r_{k+1} + r_{k+2} \approx a^{(1)} + a^{(n)} \quad (35)$$

### 3.2 $t \neq 1$

When $t$ not equal to 1. In a situation similar to two dimensions, the r value will converge to a single value relatively quickly.

for the case where $t < 1$,The system quickly reaches a balanced state after a number of iterations, and the r value will stabilize near a fixed value $r_e$ and slowly change.we have $r_e = \frac{a^{(1)}+a^{(n)}}{2t}, r_e \in (\frac{a^{(1)}+a^{(n)}}{2}, a^{(1)})$

for the case where $t > 1$,according to the analysis above, the $r$ value is no longer stable and still appear to be chaotic. However, unlike in 2 dimensions where there is only one definite single stable orbit, in higher dimensions there are several different orbits are actually narrow bands. At the beginning, the system may be in one state, and with increasing iterations, other orbital states will emerges until finally it stabilizes.there is a small amount of data outside these main orbits.As shown in Figure(3), The blue points are generated by the function $G(r)$, the orange points are generated by the function $G(r)^{-1}$, and the green points are generated by the function $Y(x) = x$.

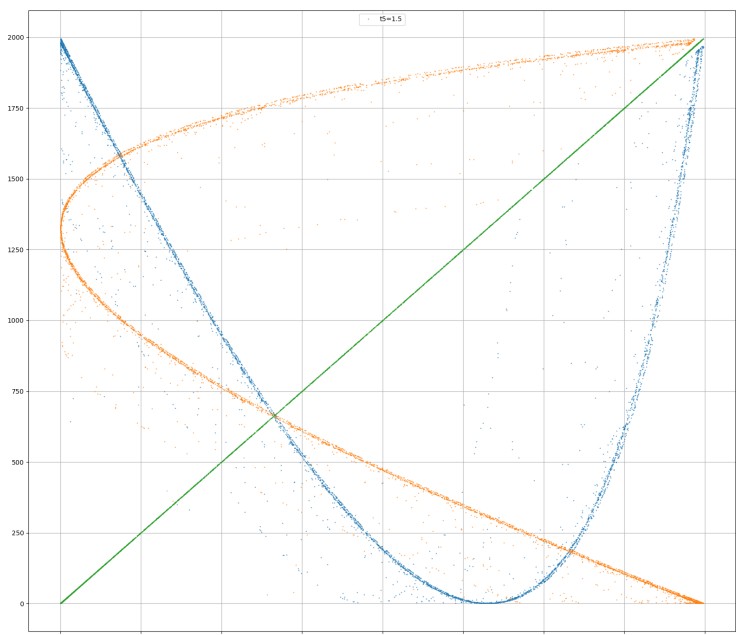

Figure 3: $G(r)(a^{(1)} = 2000, a^{(10000)} = 0.01, t = 1.5)$

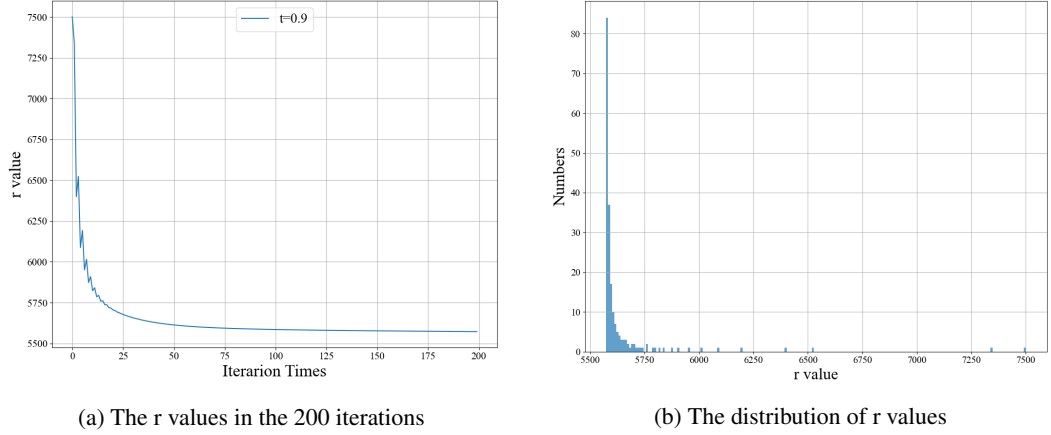

(a) The r values in the 200 iterations

(b) The distribution of r values

Figure 4: r value when $t = 0.9$

## 4 EXPRIMENT

Now, considering an example as follow

$$f(x) = \sum_{i=1}^{10000} a^{(i)} x^{(i)^2} \tag{36}$$

, where the sequence $a^{(i)}$ is arithmetic progression and $0.001 \le a^{(i)} \le 10000$, $x_0^{(i)}$ is a random number between 0 and 10000. We take the t value of three different situations and iterate 200 times. for t=0.9, as shown in Figure(4), the value of r stabilizes near a single value.

for t=1, as shown in Figure(5), the value of r quickly stabilizes near two values.

for t=1.1, as shown in Figure(6), the value of r no longer remains stable and may appear at any position. Since the value of r gradually changes near the stable point, the ratio of values near the stable point is relatively larger.

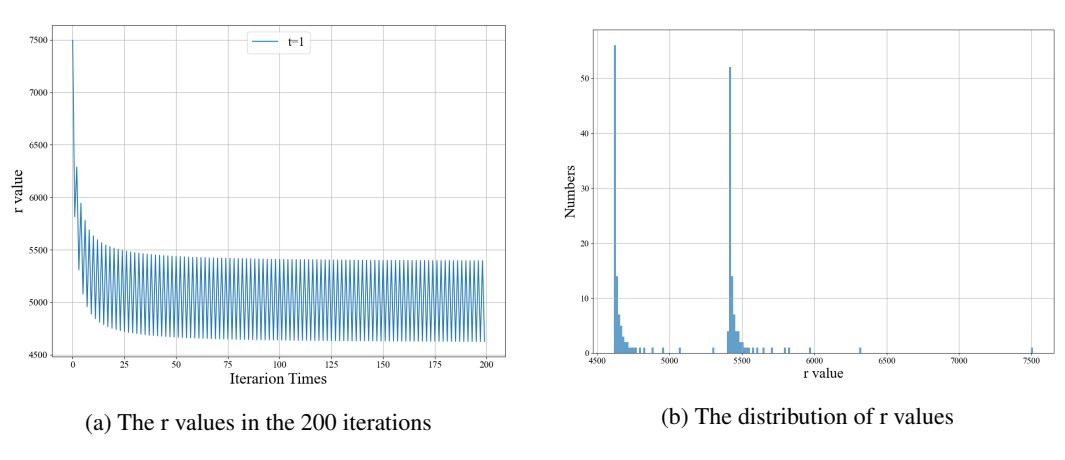

(a) The r values in the 200 iterations

(b) The distribution of r values

Figure 5: r value when $t = 1$

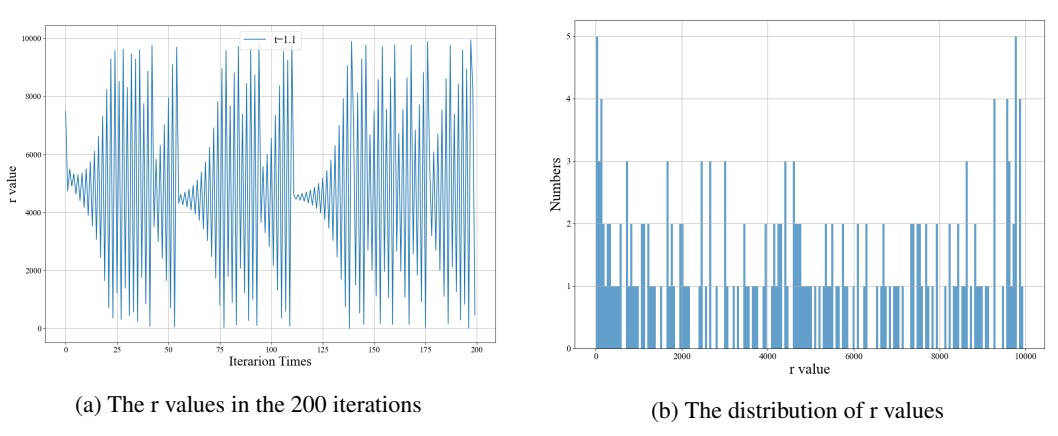

(a) The r values in the 200 iterations

(b) The distribution of r values

Figure 6: r value when $t = 1.1$

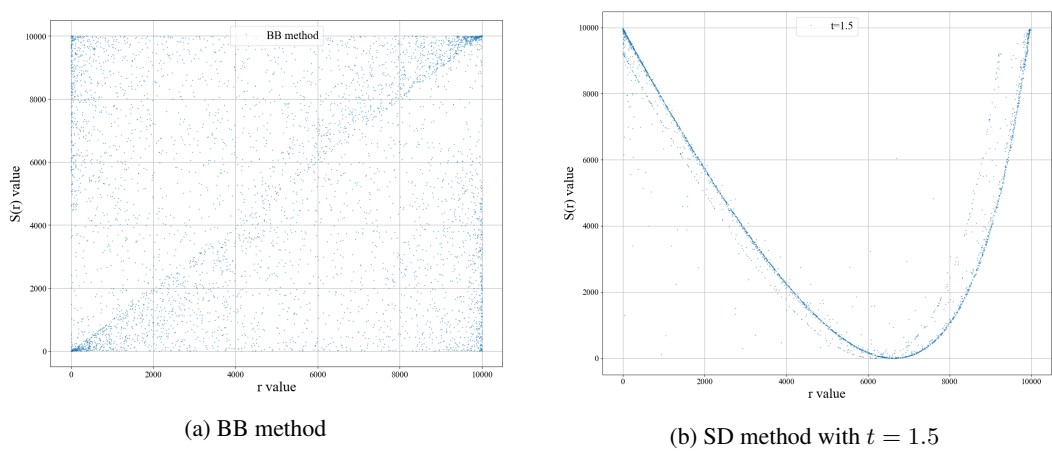

(a) BB method

(b) SD method with $t = 1.5$

Figure 7: $G(r)$ function

we further compared the $G(r)$ of the BB method and the SD method(t=1.5). as shown in Figure(7),It can be observed that the $G(r)$ of the SD method has a relatively clear trajectory, and as the number of iterations increases, the trajectory becomes more definite. On the other hand, the BB method does not have a trajectory and may fill up all the points in the space.

## 5    CONCLUSION

We analyzed the SD method by taking the reciprocal of the optimal step size $r$ and introducing a multiplicative factor $t = \frac{1}{s}$. We found that the values of $r$ before and after each iteration follow a certain pattern, which we represented using a function $G(r)$.Interestingly, this function actually describes a chaotic system. We calculated the fixed points of this system and found that, depending on the value of the multiplicative factor $t$, these fixed points correspond to different types of behavior: one type is stable with a single fixed value, another type is in a critical state with two fixed values, and the third type is unstable, causing r to jump along the main trajectory. Since the first two states correspond to fixed $r$ values and descent rates, they do not offer any advantage for the components in the direction of small eigenvalues or for overall convergence. In contrast, the unstable state allows $r$ to take on arbitrary values. Therefore, in the future, we can explore the unstable state to potentially accelerate convergence.

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
