# OpenReview forum: "An Analysis of the Cauchy Method for Different Steplength Coefficient"
_ICLR.cc/2026/Conference — Submitted to ICLR 2026_

### Official Review · Reviewer_7Vez · 2025-10-29

**Soundness:** 2
**Presentation:** 1
**Contribution:** 1
**Rating:** 0
**Confidence:** 4

**Summary:**

The paper analyzes step-size selection for convex quadratic optimization problems.

**Strengths:**

The manuscript clearly does not meet the bar for publication at ICLR:
- Significance to the machine learning community is unclear
- The optimization of convex quadratics has been studied since more than 180 years. It is unclear what the research contributes to the state of the art.
- The presentation does not meet the standards of a top-tier venue.

**Weaknesses:**

See above.

**Questions:**

None.

---

### Official Review · Reviewer_2t5Q · 2025-10-30

**Soundness:** 1
**Presentation:** 1
**Contribution:** 1
**Rating:** 0
**Confidence:** 5

**Summary:**

The paper introduces an analysis of the gradient descent method solving quadratic function, especially in two-dimensional space. The writing and presentation are very poor, lacking necessary punctuation (such as commas and periods), which are fundamental to writing a research paper. Moreover, the contribution in 2-D is insufficient to warrant publication, especially considering that this is a flagship conference for AI.

**Strengths:**

No strength.

**Weaknesses:**

Very poor writing and presentation, with missing basic punctuation.
Major contribution limited to a trivial 2-D convex quadratic case.

**Questions:**

No question.

---

### Official Review · Reviewer_pwf8 · 2025-11-01

**Soundness:** 2
**Presentation:** 1
**Contribution:** 2
**Rating:** 2
**Confidence:** 4

**Summary:**

This paper examines optimization methods, focusing specifically on the convex quadratic optimization problem. The paper identifies a phenomenon related to the reciprocal of the optimal step length. A two-dimensional case is presented to illustrate this finding.

**Strengths:**

This paper investigates how different coefficients affect the convergence of the entire system.

**Weaknesses:**

The overall paper is not well written.

Although this is a mathematical optimization paper, it should be better polished to improve readability. Symbols should be defined clearly, and more background information should be provided. Since this is an AI conference, it would also be beneficial to include experiments related to AI.

**Questions:**

see the weakness

---

### Official Review · Reviewer_V1cP · 2025-11-01

**Soundness:** 1
**Presentation:** 1
**Contribution:** 1
**Rating:** 0
**Confidence:** 4

**Summary:**

This work aims to provide new results regarding steepest descent on quadratic problems. Most importantly, the authors propose an interesting approach of analyzing how the method evolves with respect to the stepsize, instead of the traditional analyses focusing on the iterates.

**Strengths:**

This work offers a clear alternative viewpoint by modeling the steepest descent method as a one-dimensional dynamical system in terms of the optimal step size. For the two-dimensional quadratic case, the authors provide a concise and explicit characterization of the corresponding dynamics, which is then extended to the general multidimensional setting. The paper also includes experimental results that qualitatively support the analytical findings.

**Weaknesses:**

The scope of this work is extremely narrow.
Not only does it focus on the steepest descent method applied to **quadratic** problems, but the theoretically sound results are limited to the two-dimensional case, with claims for higher-dimensional extensions lacking formal justification.
Consequently, the connection to modern machine learning optimization problems is very weak, raising questions about whether this paper fits the scope of the intended venue.
Moreover, while the paper does not propose a fundamentally new optimization method but rather offers an alternative perspective on an existing algorithm, its limited scope makes it difficult to argue that it provides substantial new insight.

Also, the overall presentation is not up to publication standards. The paper contains numerous grammatical and stylistic issues (most notably, uncapitalized sentence beginnings), inconsistent notation without proper explanation or derivation (for example, the definition of $r$ in Equation 9), incorrect cross-references (such as “minimizing Eq.(3)” around line 034), and the use of unjustified or imprecise terminology (for instance, describing the system as “chaotic”, which is not supported by a rigorous definition or analysis). These issues collectively make the paper difficult to follow and detract from its technical clarity.

**Questions:**

1. What is the ultimate goal of this analysis? As early as 2015, Kalousek already showed that step sizes can even be randomly chosen, and more recent works, such as the Silver Stepsize Schedule by Altschuler and Parrilo (2025), demonstrate that occasionally using large step sizes can actually accelerate convergence. These results suggest that one cannot meaningfully characterize or predict the convergence of the iterates $x_k$, which are what ultimately matter, solely by analyzing the evolution of the step sizes. How does the proposed one-dimensional analysis in terms of  $r_k$ advance our understanding of the underlying optimization process beyond what is already known from these prior studies?

2. Could the authors clarify whether they envision the proposed analysis as a tool for designing new step-size adaptation schemes, or purely as a qualitative study?

---

### Meta-Review · Area_Chair_WtLL · 2026-01-03

**Summary:**

This paper considers the analysis of stepsize for convex quadratic optimization problems. All reviewers agree that the paper is of poor quality and readability. I think it should be **rejected**.

**Reviewer Concerns:**

There is no rebuttal.

**Reviewer Scores:**

I think the reviewers would not change their score if they had been able to participate fully in the discussion.

---

### Decision · Program_Chairs · 2026-01-26

Reject